# Child- and family-specific communication skills trainings for healthcare professionals caring for families with parental cancer: A systematic review

**Wiebke Frerichs**[1]*, **Wiebke Geertz**[1], **Lene Marie Johannsen**[1], **Laura Inhestern**[1], **Corinna Bergelt**[1,2]

**1** Department of Medical Psychology, University Medical Center Hamburg-Eppendorf (UKE), Hamburg, Germany, **2** Department of Medical Psychology, University Medicine Greifswald, Greifswald, Germany

\* w.frerichs@uke.de

## Abstract

**Data Availability Statement:** There are no additional data available.

### Introduction

As part of high-quality cancer care, healthcare professionals (HCPs) play a significant role in identifying and addressing specific needs of cancer patients parenting minor children. However, HCPs experience various barriers to adequately support parents with cancer. This systematic review explores current CSTs incorporating child- and family- specific modules for HCPs in oncology. Moreover, outcome measures and effectiveness of trainings are systematically investigated.

### Methods

The systematic review was registered within PROSPERO (registration code: CRD42020139783). Systematic searches were performed in four databases (PubMed, Cinahl, PsycInfo, Web of Science) in 12/2020, including an update in 12/2021 and 08/2022. Quantitative, primary studies fulfilling the pre-defined inclusion criteria were included. Due to the expected heterogeneity a meta-analysis was not conducted. Study selection and quality assessment were conducted by two independent researchers, data extraction by one. Study quality was assessed using an adapted version of the National Institutes of Health quality assessment tool for pre-post studies without control group.

### Results

Nine studies were included in this review following an experimental pre-post design only. Two CSTs were specifically designed to improve communication with cancer patients parenting minor children, the remaining seven incorporated a brief family module only. Seven programs were face-to-face trainings, one an e-learning and one a webinar. Eight studies found at least one statistically significant improvement in communication after training. However, quality of most studies was fair.

**Funding:** The author(s) received no specific funding for this work.

**Competing interests:** The authors have declared that no competing interests exist.

**Abbreviations:** COMFORT, COMFORT TM SM Communication Curriculum; COMSKIL, COMSKIL training program; CST, Communication skills training; GHQ, General Health Questionnaire 28; HCPs, Healthcare professionals; MBI, Maslach Burnout Inventory; NIH, National Institutes of Health.

## Conclusion

This is the first review exploring specific CSTs for HCPs caring for cancer patients parenting minor children. As only two CSTs focused on parental cancer, evidence on the effectiveness of such CSTs is limited. Existing CSTs should be evaluated properly and include details on content of family modules. Further studies including and evaluating specific CSTs focusing on parental cancer are needed in order to strengthen HCPs' competencies to meet specific needs of patients parenting minor children.

## Introduction

Approximately 14–25% parents with dependent children are diagnosed with cancer [1–3] which can have a major impact on the entire family. Cancer patients parenting minor children experience increased levels of stress and anxiety compared to patients without minor children [4, 5]. Additional to the burden of the life-limiting disease and its treatment, parents with cancer worry about how to maintain family life and their role as a "good" parent and supporter [6–9]. Parents often feel insecure if, when and how to communicate with their children about cancer and how to adequately address their children's needs [8, 10, 11].

Children of parents having cancer experience major challenges in their family routine and increased psychosocial stress [6, 12]. Even without knowing, they feel that something serious is going on [13]. Providing age-appropriate information and timely communication about parental cancer can decrease the risk of developing negative psychological and physical consequences in affected children [12, 14].

Healthcare professionals (HCPs) have a significant role in identifying patients parenting minor children, their specific needs and—if necessary—initiating supportive, psychosocial care [1, 6, 8, 11]. In order to provide high-quality, patient-centred cancer care, involvement of family and their specific needs is essential [15, 16]. Family members are often the primary support for cancer patients [17] and act as caregiver and thus are impacted by cancer as well [18]. As family communication is associated with relationship functioning and adjustment to the cancer diagnosis [18], it is essential for HCPs to provide support to cancer patients and their families on family communication issues, e.g., open communication. In order to identify potential cancer patients parenting minor children, it is key to know about the patient's family status and if applicable to proactively address child- and family specific themes within cancer care. Previous studies show that parents with cancer wish for support and guidance from their HCPs about child- and family-specific aspects, especially on communication with their children [1, 9, 10, 19]. However, current results show that less than 50% of HCPs routinely communicate about child- and family-specific themes with their patients [20]. Barriers of HCPs to include child- and family-specific aspects routinely in cancer care are e.g., lack of specific competencies and knowledge as well as time pressure or structural barriers [8, 11, 21, 22]. Additionally, other studies report that HCPs feel insufficiently trained in providing basic adequate psychosocial support to cancer patients parenting minor children [11, 23].

In order to address these major barriers in HCP's communication about child- and family-specific aspects, adequate trainings are needed to improve communication skills and competencies for HCPs in oncology [10, 24].

Over the last decade, various communication skills trainings (CSTs) have been developed and implemented to improve communication skills in oncology. Findings indicate

improvements in HCP's communication skills, namely increasing empathy [25], knowledge and self-efficacy [26] or in certain patient-reported outcomes, e.g., patient satisfaction [26].

Considering the described relevance and specific burden of affected parents, CSTs should also address these aspects. However, despite many CSTs being developed for HCPs in oncology in recent years [25–27], it remains unclear whether and to what extent child- and family-related aspects are addressed in these CSTs and previous reviews on CSTs have not included this topic [25].

To close this gap, this systematic review aims to a.) provide an overview of existing CSTs for HCPs working in oncology addressing child- and parent-specific aspects in cancer care, b.) explore reported outcome measures associated with the CSTs and c.) gather existing evidence of effectiveness of these trainings.

## Materials and methods

The systematic review was registered in the International Prospective Register of Systematic Reviews (PROSPERO, registration code: CRD42020139783) and follows the updated guideline for reporting systematic reviews (PRISMA 2020 statement [28]).

### Data sources and search strategy

An electronic literature search was performed in the databases of PubMed, Cinahl, PsycInfo and Web of Science with no limitation regarding the publication year. The search was conducted on December 9$^{th}$, 2020, was developed in PubMed and adapted to the other databases. A search update was conducted on December 3$^{rd}$, 2021 and on August 12$^{th}$ 2022. A librarian of the Central Medical Library Hamburg was consulted to review the final search strategy.

The systematic search strategy consisted of a combination of different terms and keywords from the following four domains: (i) communication skills training, (ii) healthcare professional, (iii) oncology, and (iv) parent/family (see Table 1).

Articles on pediatric oncology as well as qualitative studies were excluded. Our primary electronic search strategy was complemented by a hand search, consisting of citation tracking of included articles.

### Eligibility criteria and study selection

Due to language restriction of the authors, peer-reviewed publications in English or German were retrieved. We included studies reporting any type of CST with a pre-post design (e.g., single arm intervention studies or studies including a control group) regarding outcomes assessing change of communication competencies, comprising at least one module on child- or parent-specific aspects in cancer care for HCPs caring for adult cancer patients. The applied in- and exclusion criteria are displayed in Table 2. However, despite our extensive search strategy only two studies were identified during the study selection process to focus on child- and parent-specific aspects within their CSTs. Therefore, we decided to broaden the focus of this systematic review and to include studies, which entail a child- and family-specific module within their CST.

To manage and facilitate the selection process, search results were imported into the reference management software EndNote (Version EndNote X9.3.2) and duplicates were removed. One author (WF) conducted the title and abstract screening. All potentially relevant articles according to the defined inclusion and exclusion criteria were included for full text screening. Full texts were independently assessed for eligibility by two reviewers (WF, WG). Disagreement between reviewers was resolved by discussion; where necessary, a third reviewer (LI) was consulted.

**Table 1. Search strategy exemplary for the database Pubmed, adjusted according to other electronic databases.**

| | Search Strategy |
|---|---|
| #1 | communication[Title/Abstract] OR "talking"[Title/Abstract] OR "talk*"[Title/Abstract] OR "competenc*"[Title/Abstract] OR "skill*"[Title/Abstract] OR "consultat*"[Title/Abstract] OR "communication"[MeSH Major Topic] OR "education, medical, continuing"[MeSH Major Topic] OR "education, nursing, continuing"[MeSH Major Topic] |
| #2 | train*[Title/Abstract] OR training[Title/Abstract] OR education*[Title/Abstract] OR seminar*[Title/Abstract] OR program*[Title/Abstract] OR teach*[Title/Abstract] OR workshop*[Title/Abstract] OR cours*[Title/Abstract] OR develop*[Title/Abstract] OR intervention*[Title/Abstract] OR manual*[Title/Abstract] OR plan*[Title/Abstract] OR instruction*[Title/Abstract] OR curriculum*[Title/Abstract] OR e-learning[Title/Abstract] OR electronic learning[Title/Abstract] OR online[Title/Abstract] OR web-based[Title/Abstract] OR webbased[Title/Abstract] OR tool*[Title/Abstract] |
| #3 | (#1 AND #2) OR "communication skill* training*" |
| #4 | (health[Title/Abstract] OR healthcare[Title/Abstract] OR health care[Title/Abstract]) AND (personnel[Title/Abstract] OR staff[Title/Abstract] OR provider*[Title/Abstract] OR professional*[Title/Abstract]) OR oncologist*[Title/Abstract] OR oncolog*[Title/Abstract] OR nurs*[Title/Abstract] OR doctor*[Title/Abstract] OR practitioner*[Title/Abstract] OR medical[Title/Abstract] OR psychologist*[Title/Abstract] OR psychological[Title/Abstract] OR social work*[Title/Abstract] OR "health personnel"[MeSH Major Topic] |
| #5 | "parenting"[MeSH Major Topic] OR "parent*"[Title/Abstract] OR "mom*"[Title/Abstract] OR "dad*"[Title/Abstract] OR "mother*"[Title/Abstract] OR "father*"[Title/Abstract] OR "famil*"[Title/Abstract] OR "matern*"[Title/Abstract] OR "patern*"[Title/Abstract] |
| #6 | ("medical oncology"[MeSH Major Topic]) OR ("oncology nursing"[MeSH Major Topic]) OR ("cancer survivors"[MeSH Major Topic]) OR ("cancer"[Title/Abstract]) OR (oncolog*[Title/Abstract]) OR (tumor*[Title/Abstract]) OR ("tumour"[Title/Abstract]) OR ("leukeamia"[Title/Abstract]) OR (palliativ*[Title/Abstract]) OR (metastat*[Title/Abstract]) OR (malign*[Title/Abstract]) |
| #7 | #3 AND #4 AND #5 AND #6 |
| #8 | #7 NOT (pediatric[Title/Abstract]) |
| #9 | #8 NOT (qualitative[Title]) |

**Table 2. Inclusion and exclusion criteria.**

| Inclusion | Exclusion |
|---|---|
| **Study type and design** | |
| Quantitative intervention studies | Qualitative studies, observational studies without intervention, studies without a pre-post measurement of outcomes |
| | Study is not cancer specific (e.g., mental illness) |
| | Dissertations, conference abstracts etc. |
| **Participants** | |
| Healthcare professionals (HCPs) working in oncology or with cancer patients (e.g., nurses, physicians) | HCPs not working with cancer patients or not having any experience caring for cancer patients |
| HCPs within healthcare educational programmes (e.g., nursing or medical students); | HCPs working in paediatrics |
| HCPs working in all settings (e.g., outpatient, inpatient) | |
| **Interventions** | |
| Studies evaluating a communication training or educational program including at least a module on child-, parent- or family-specific themes; | Intervention is not a communication skills training or is not an educational intervention to improve HCPs' communication with cancer patients |
| Intervention aims to improve healthcare professional's communication skills, behavior or knowledge | Intervention aimed to improve communication within paediatric oncology |
| **Date** | |
| No restrictions | |
| **Language** | |
| English or German | |
| **Availability** | |
| Fulltext needs to be available (e.g., contact by e-mail to corresponding author) | |
| **Study must be published in a peer-reviewed journal** | |

## Data extraction and synthesis

As we were expecting a large heterogeneity of studies including a large variation in participants, outcome measures or type of CST being used, we synthesized findings of the included studies in the form of a narrative review. A data extraction form was developed including the following information: aims/background, study design and methods; details of CST (e.g., development, setting, duration, content, teaching strategies); details on child- and family-specific module; characteristics of participants; CST outcome measures and results. The form was independently pilot tested by two reviewers (WF, WG) with one randomly selected study included in this review. Data extraction of included studies was systematically performed by one reviewer (WF), final results were discussed with two other reviewers (WG, LI).

Intervention outcomes and findings were categorized based on Kirkpatrick's framework for training evaluation based on the following levels: 1. Reaction–Participant's satisfaction with the training; 2. Learning–Participant's change of attitudes, increase in knowledge and skills; 3. Behavior–Participant's change in behavior; 4. Results–other improvements in patient-oriented healthcare (e.g., participants well-being) [29].

## Quality assessment

Methodological quality of included studies was independently assessed by two reviewers (WF, WG) using a slightly modified version of the National Institutes of Health (NIH) quality assessment tool for pre-post studies without control group [30]. This tool was selected as all included studies were quasi-experimental studies with a pre-post design. none including a control group. Study quality could be rated as good, fair or poor. Any disagreement between reviewers was resolved by discussion and, where necessary, a third reviewer (LI) was consulted.

# Results

The main literature search identified two studies specifically addressed the subject of cancer patients parenting minor children within their CST and five studies incorporated a brief family module within their CST. The first update added another two studies evaluating a CST for HCPs in oncology, including a brief module on family-specific aspects in cancer care. In total, nine studies were included in this review (Fig 1).

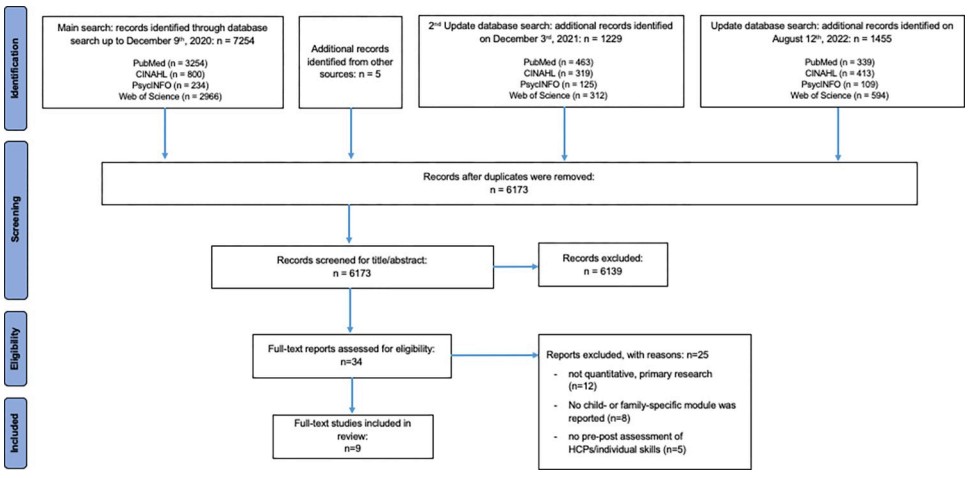

**Fig 1. PRISMA flow diagram for systematic reviews.**

## Description of included studies

Table 3 gives an overview of included studies. All included studies were published between 2008 and 2021. Five studies were conducted in the North America [31–35], two in Australia [23, 36], one in Africa [37] and one in Europe [38]. Included studies used a quasi-experimental design with pre-post measurement only and no studies were identified including a control-group.

## Participants

Most studies included qualified HCPs [23, 31, 33–36, 38], two studies included nursing students only [32, 35] (see Table 3). In total, 1578 HCPs participated in the included studies. Six studies including nursing professionals only [23, 31–35], three studies including HCPs of various disciplines (e.g., nurses, doctors, social workers) [36–38], in which there was a high proportion of nurses (e.g. [38]). In two studies only female HCPs participated [33, 34], four other studies included mainly female participants (range 75–97% [23, 32, 35, 36]). In studies reporting on mean age of participants, mean age ranged from 24 to 47 years [23, 32, 34, 35]. Professional experience varied from overall working experience [37] to working in oncological setting [23, 33, 38] or in palliative care [34]. Two studies assessed previous communication skills training participation [34, 38]. One study assessed if participants had currently a serious illness in family member (34%) and previous history of bereavement of a first degree relative (51%) [23].

## CST characteristics

Of the nine included studies, one training was an e-learning training [35] and one a webinar series [37]. The remaining CSTs were face-to-face trainings [23, 31–34, 36, 38]. The duration of the programs varied substantially in length, ranging from 30–40 minutes [38] to a 2-day program [36]. Group size of trainings varied between studies, with small groups of n = 3–8 participants each [23, 33, 34, 36, 38] and large groups of e.g., up to n = 158 participants per training [36]. The content of the CSTs was either developed based on a literature review [23, 31, 35, 36, 38], a needs assessment (e.g., focus group or survey [23, 31, 32]) or input through a workshop with experts [23, 35, 36]. One study reported on pilot-testing their intervention [35].

Detailed description of the CSTs, outcome measurements and results of the included studies are presented in Table 4.

## Content and development of the CSTs including a child- or family-specific module

Of the nine studies included, only two included a CST for HCPs specifically addressing the subject of cancer patients parenting minor children [23, 38]. The remaining seven studies incorporated a brief family module within their CST. This brief family module often entailed themes e.g., how to communicate with families of cancer patients [31, 32, 35–37] or how to involve the family in cancer care [33, 34]. Detailed information if and in which way the family modules refer to children as relatives in particular or if parental issues were covered was not reported within the studies.

Two studies [31, 32] applied the COMSKIL training program [39] and four [33–35, 37] the original or an adapted version of COMFORT [TM][SM] Communication Curriculum [40], a CST specifically designed for nurses. Three studies developed their own CST program [23, 36, 38]. The description of the family module differed slightly between studies using the original

**Table 3. Characteristics of the included studies.**

| Author, Year Title | Country | Design | Name of CST | Child- or family-specific module | Participants (HCPs) | N of participants | Participants characteristics | CST duration, setting, group size, number of sessions | Quality assessment NIH score |
|---|---|---|---|---|---|---|---|---|---|
| **Banerjee et al., 2017** The implementation and evaluation of a communication skills training program for oncology nurses | USA | Quasi-experimental Pre-Post Evaluation | adapted COMSKIL | Responding to challenging interactions with families (one of three modules) | Oncology nurses from various inpatient settings (e.g., acute care, pediatrics) at MSKCC | 342 | N/R | 1-day Face to face ≈12 nurses per training | poor |
| **Cannity et al., 2021** Acceptability and efficacy of a communication skills training for nursing students: Building empathy and discussing complex situations | USA | Single-arm pre-post design | adapted COMSKIL | Responding to challenging interactions with families (one of three modules) | Nursing students from a large cancer center in the northeastern United States; as part of their clinical education during their final year of bachelor nursing education | 158 | 87% female Average age 23.7 years (SD = 3.70) 84% white/Caucasian 9% Latino 4% Asian/ Asian.-American 1% black/African-American | 1-day Face-to-face Participants per training = NR | fair |
| **Cronin & Finn, 2017** Implementing and Evaluating the COMFORT Communication in Palliative Care Curriculum for Oncology Nurses | USA | Quasi-experimental Pre-Post Evaluation | adapted COMFORT | Module F: incorporating family into plans of care, understanding characteristics of various caregivers | Nurses from 2 oncology units in palliative care | 20 | 100% female Age: 60% of 21–29 years; 90% white/ Caucasian, 10% African American; Professional experience: as practicing nurse 0–5 years = 80%, as oncology nurse 0–5 years = 85%; Highest level of education: 5% associate degree, 70% bachelor's degree, 25% master's degree; | 4 hours Face-to-face ≈5 nurses per training | fair |
| **Fuoto & Turner, 2009** [34] Palliative Care Nursing Communication: An Evaluation of the COMFORT Model | USA | Quasi-experimental Pre-Post Design | COMFORT | Family: support caregiver involvement and understanding | Nurses from one palliative care unit in a large tertiary care center | 14 | 100% female Age: M 46.9 years (SD 11.9) Employment as registered nurse in years: mean 16.5 (SD 9.9), in palliative care: mean 3.2 (SD 4.4); highest level of education: 36% associate degree, 50% bachelor's degree, 14% master's degree; 100% of HCPs reported on at least 1 previous CST | 4 hours Face-to-face ≈4–6 nurses per training | fair |

*(Continued)*

Table 3. (Continued)

| Author, Year / Title | Country | Design | Name of CST | Child- or family-specific module | Participants (HCPs) | N of participants | Participants characteristics | CST duration, setting, group size, number of sessions | Quality assessment NIH score |
|---|---|---|---|---|---|---|---|---|---|
| **Quinn et al., 2008** "Palliative care: the essentials": evaluation of a multidisciplinary education program | Australia | Quasi-experimental Pre-Post Design | "Palliative Care: The Essentials" | Session IV: Family-centered care Session VI: Communicating with patients, families and colleagues Session IV: Care of an imminently dying person and their family | All HCPs with university degree (or equivalent) | 495 (Nurses = 402, allied HCPs = 44, doctors = 8, other staff = 41) | 96% female Age: 60% between 40–59 years | 2-day program Face-to-face 4 trainings of 92–158 HCPs each | fair |
| **Semple et al., 2017** How education on managing parental cancer can improve family communication | Northern Ireland | Quasi-experimental Pre-Post Design | Educational intervention for HCPs to communicate with parents diagnosed with cancer | Specific CST to communicate with parents diagnosed with cancer | Oncology professionals | 259 in total (registered nurses = 147, allied HCPs = 60, social worker = 17, medical staff = 5, Other = 30) | Range of years working with cancer patients: 39% 0–5 years 12% 6–11 years 27% 11–20 years 14% 20+ years 8% missing data Received previous training in supporting families with parental cancer: 9% | 30–45 min. session Face-to-face 35 trainings of 3–5 participants | fair |
| **Turner et al., 2009** Enhancing the capacity of oncology nurses to provide supportive care for parents with advanced cancer: evaluation of an educational intervention | Australia | Quasi-experimental Pre-Post Design | Educational intervention with self-directed learning manual + Communication skills training workshop | Specific CST to provide supportive care for patients with advanced cancer who have dependent children | Oncology Nurses | 32 | 94% female Age in years: Mean 39.7 (SD 10.4) Marital status: 60% married, 22% single, 9% divorced; Years in oncology: Median 9 (range 0.5–25) Current serious illness in family member: 34% Experiences bereavement of first degree relative: 51% | 1-day face-to-face 5 trainings of 5–8 participants | good |

(Continued)

Table 3. (Continued)

| Author, Year Title | Country | Design | Name of CST | Child- or family-specific module | Participants (HCPs) | N of participants | Participants characteristics | CST duration, setting, group size, number of sessions | Quality assessment NIH score |
|---|---|---|---|---|---|---|---|---|---|
| **Wittenberg et al., 2020**<br><br>Caring for Family Caregivers: a Pilot Test of an Online COMFORT™ SM Communication Training Module for Undergraduate Nursing Students | USA | Quasi-experimental Pre-Post Design | adapted Module F of COMFORT | Adapted Family module: e.g., Family Communication Patterns, the Four Family Caregiver Types and Strategies for communicating effectively with each type | Undergraduate bachelorette students | 128 | 75.6% female Mean age in year: 24.51<br><br>75.6% female<br><br>31.5% Caucasian, 13.4%, multi-racial, 12.6% African-American<br><br>English-speaking: 82.7%<br><br>1st year students: 7.1%<br><br>2nd year students: 22%<br><br>3rd year students: 40.9%<br><br>4th year students: 18.1% | 1-hour online course Participants per training = NR | fair |
| **Wittenberg et al., 2021** [37]<br><br>Sharing COMFORT Communication Training With Healthcare Professionals in Nairobi, Kenya: A Pilot Webinar Series | Kenya | Quasi-experimental Pre-Post Design | Adapted COMFORT curriculum for a Webinar-Based Training including the 3 of COMFORT's modules: C-Connect, O-Options, F-Family caregivers; | Adapted Family module with following objectives: define family communication patterns; identify differences among family caregiver types; demonstrate awareness of family caregiver communication patterns; | HCPs in Nairobi, Kenya | 94 in pre-training survey only (physicians = 52, nurses = 24, other HCPs = 18) 12 completed 3-part webinar series and post-survey (physicians = 6, nurses = 4, other = 2) | Working experience = on average 10 years 88.3% reported having daily communication with patients, 5.3% communicated 3x a week, 2.1% only weekly with patients, 4.3% communicated less than weekly | 3x 1-hour webinar series | poor |

CST–Communication skills training; HCPs–Healthcare professionals; NIH score–National Institutes of Health (NIH) quality assessment tool for pre-post studies without control group; USA–United States of America; COMSKIL program was developed by Kissane et al. at the Memorial-Sloan-Kettering Cancer Center (MSKCC) in the USA; NR–Not Reported; COMFORT Communication Curriculum developed by Elaine Wittenberg, PhD and Joy Goldsmith, PhD originally as a 2-day face-to-face training for the palliative care setting.

**Table 4. Characteristics of communication skills training interventions.**

| Author(s), year | Development/Background & pilot-tested | Content | Teaching strategies | Evaluation (Outcomes, measures and time points) | Main results* |
|---|---|---|---|---|---|
| Banerjee et al, 2017 | • A curriculum development committee identified three CST topics for nurses (communication of empathy with patients and family, discussion of death, dying, and end-of-life goals of care; navigating through difficult family interactions); • qualitative online survey for nurses assessing common communication challenges in the previous defined CST topics • following literature and conceptual and methodological approach of the Comskil conceptual model and Comskil Training programme for oncologists Pilot-tested: NR | *Comskil training program* Introductory lecture; *Three 2-h modules*, each including: • a 30-min presentation on theoretical background, recommended communication approach and video display with simulated patient • 90-min small group role play exercises with simulated patients (3 participants) Content of the modules: 1. Responding empathically to patients 2. Discussing death, dying and end-of-life goals of care 3. Responding to challenging interactions with families Additional material: Each participant received printed booklet on each module | Presentation, Videos incl. simulated patients, written booklet for each module; facilitator-led small group role play sessions with simulated patients incl. reflections and feedback sessions from peers and facilitator, video playback; Facilitators and their professional background: Role-play sessions: facilitated by communication skills specialists (faculty, interventionists, and researches specialized in CSTs) and a specialty-specific nurse (also trained in facilitation). For lectures and presentations = NR | a.) *Program evaluation* with a paper-pencil survey assessing • post-training attitudes regarding learned skills and application (6 items, 5-point Likert scale); • effectiveness of curricula activities in each module including didactic teaching, videos, role play exercises (4 items, 3-point Likert scale); b.) *Participant learning* with • Self-reported self-efficacy retrospective pre-post measure (2 questions per module) • demonstration of skills pre- versus post-training SPAs using an 8-min video recorded interaction between nurse and a simulated patient Follow-up: none Development and Evaluation: based on Kirkpatrick model, adapted to training model; assumption of self-developed tools; | a.) *Program evaluation:* • Training was rated favorably, 90% of nurses agreed or strongly agreed with 5 of 6 learning items; • >80% rated each module as aiding in learning; b.) *Participant learning:* • Overall participants' self-reported self-efficacy significantly improved between retrospective pre- and post-training assessment [p < .001], specifically in responding empathically [p < .001], discussing death and dying and end-of-life goals of care [p < .001], and responding to challenging interactions with families [p < .001]; • demonstrated significant improvements over all skills in pre- and post-training assessment [p < .001], specifically in several empathetic communication skills [p < .001], a skill on clarifying [p < .001]s, a skill on summarizing [p < .05]. No significant change was observed in the overall areas of agenda setting skills, information organization skills or checking skills. |
| Cannity et al., 2021 | • based on needs assessment of communication challenges for oncology nurses (Banjeree et al 2016) • additional consultation with experts in the field of nursing and communication Pilot-tested: NR | *Comskil training program* One day, three-topic workshop. *Three modules*, each including: • a 30-min presentation on theoretical background, recommended communication approach and video display with simulated patient • 90-min small group role play exercises with simulated patients and clinical vignettes (3 participants), if possible, based on workplace setting (e.g. urgent care) Content of the 3 modules: 1. Responding empathically to patients 2. Discussing death, dying and end-of-life goals of care 3. Responding to challenging interactions with families Each participant received printed booklet on each module | Presentation with goals of each module, specific techniques demonstrating current literature on the skills, videos demonstrating techniques, role play incl. simulated patients, video-feedback and pre-post SPAs; written booklet for each module; Facilitators and their professional background: Role-play sessions: facilitated by communication skills specialists (faculty, interventionists, and researches specialized in CSTs) and a specialty-specific nurse (also trained in facilitation). For lectures and presentations = NR | a.) Perception of course effectiveness with a paper-pencil survey assessing • Self-reported effectiveness and applicability of skills taught (8 items, 5-point Likert scale); • Rating of specific components aided learning (didactic teaching, videos, role play exercises, 3 items, 3-point Likert scale); b.) Participant learning with • Self-reported pre- and post-training confidence in the specific communication skills (1 pre-, 2 post-training items, 5-point Likert scale); • demonstration of skills pre- versus post-training SPAs using an 8-min video recorded interaction between student and a simulated patient including clinical vignettes developed by nursing experts; coding by two independent blinded raters with Comskil coding system developed by Bylund and colleagues (2011). Follow-up: none Development and Evaluation: based on Kirkpatrick model, but as efficacy study focus on level 1 (reaction of participants to program) and level 2 (evaluation of participants' learning) | a.) *Course effectiveness:* • 90% endorsing engagement or strong agreement with course efficacy overall; • 90% agreed or strongly agreed module 1 and 2 were useful • For module 2, 90% found skills learned useful • 90% of participants reported techniques applied somewhat or strongly helpful in learning; b.) *Participant learning* • Overall participants' confidence significantly improved between pre- and post-training assessment across all three domains [p < .01]; Communication skill usage significantly increased between pre- and post-training [p < .001]; |

(Continued)

**Table 4.** (Continued)

| Author(s), year | Development/Background & pilot-tested | Content | Teaching strategies | Evaluation (Outcomes, measures and time points) | Main results[a] |
|---|---|---|---|---|---|
| Cronin & Finn, 2017 | Synthesized COMFORT communication curriculum from Wittenberg-Lyles et al. 2010, 2014; not further specified. Pilot-tested: NR | COMFORT: C communication: Narrative clinical practice, task and relational communication O orientation and opportunity: Health literacy, cultural support M mindfulness: Remaining present in the moment, adapting to change; F family; incorporating family into plans of care, understanding characteristics of various caregivers O openings: Recognizing opportunities for discussion, engagement in moments of stress/tension R relating: Exploring uncertainty and building trust with patient/families T team: Facilitating team meetings, improving communication and interdisciplinary collaboration 'Taught by 2 instructors, with introduction and description of each of the 7 COMFORT components followed by questions and clarification (approx. 2 hours); intermittently using role playing and videos, exercise on plain language (approx. 1 hour total); discussions and exchange (1 hour); | Power-point lectures describing each of the COMFORT components; videos and role-playing to enhance learning; firsthand videos accounts from an oncology patient sharing his interactions and experiences with viewers; small-group discussions to share personal stories and experiences; exercises on importance of plain language; Facilitators and their professional background: 2 nursing oncology practice specialists | *Precourse-postcourse surveys (2 weeks after attending course)* *Program Evaluation:* quality of the program, satisfaction of learners and whether objectives met; *Participant learning with pre-post survey:* a.) Communication Skills Attitude Scale (CSAS, 26 items, 5-point Likert scale) assessing attitudes toward learning communication skills; b.) Perceived Importance of Medical Communication (PIMC, 12 items, 5-point Likert scale) assessing attitudes towards importance of medical education adapted to fit nursing; c.) Caring Efficacy Scale (CES, 30 items, 6-point Likert scale), assessing nurses' self-perception of competency in nursing practice; Follow-up: none Development: Mix of validated questionnaires (for participants learning, slightly adapted to fit nursing) and self-developed questions for program evaluation; | *Program Evaluation:* overall positive response, some preferred longer course to continue dialogues and discussions regarding their experiences communicating with patients, families and teams. *Participant learning* No statistically significant changes in pre- to post-course measurement, but majority of results (64%) indicate increase of mean scores: a.) CSAS: 86% improvement in nurses' attitudes toward learning communication skills; b.) PIMC: 75% improvement in perceived importance c.) CES: 43% improvement in perceived competencies |
| Fuoto & Turner, 2019 | Synthesized COMFORT communication curriculum from Wittenberg-Lyles et al. 2010, 2015; not further specified. Pilot-tested: NR | COMFORT principles; within role play exercises each nurse was given the opportunity to be the patient, family and the nurse; after each role play a debriefing was held to discuss lessons learned and reinforce how to use COMFORT to guide communication | Lecture and group discussions; role play exercises with peer group feedback and discussions afterwards; COMFORT card for badge holder and posters in staff break rooms and bathrooms; Facilitators and their professional background: NR | Between subject's pre-post design 3 months before implementation and 3 months after implementation; a.) *Demographic survey* (6 items) b.) *Communication confidence survey* (5 items, 4-point Likert scale) c.) *Communication satisfaction survey*– measuring perceived satisfaction during experiences of EOL communication (4 items, 5-point Likert scale) d.) *2 short answer questions* (self-developed) only at 3-month follow-up to measure *changes in daily practice and communication* since participation; e.) *Patient-family satisfaction with EOL care* (FEPC, 39 question survey but only 18 used for the study)–a post death survey 6-weeks after the death of a patient; Follow-up: 3-months post participation only for d.) Development: two tools (b, c) developed by others, but not validated; one tool validated, but adjusted questionnaires (FEPC), one self-developed tool (d); | b.) *Communication confidence:* statistically significant increases in confidence in overall ability to communicate in difficult conversations across the three time points [p = .002]; in competence communicating with families in crisis [p < .001], in competence managing emotional needs at EOL [p < .001], competence managing conflicts [p < .001] and in overall competence communicating in difficult EOL situations [p < .001]; c.) *Communication satisfaction:* statistically significant increases in satisfaction with overall ability to communicate in difficult conversations across the three time points [p < .001]; in satisfaction managing emotional needs at EOL [p < .001], in managing conflict and in overall satisfaction communicating in difficult EOL situations [p < .001]. d.) *changes in daily practice:* three themes emerged on COMFORT impact on daily practice: increased confidence, preparation and increased awareness; *changes in communication practice:* six themes = increased confidence, communication, mindfulness, family, orientation, and no change; most reported using the model regularly, some reported it existed already in their practice; e.) *Patient-family satisfaction with EOL care* (n = 50 respondents, 23 precourse, 27 postcourse); no significant difference in score before versus after training, although baseline results were generally high and most respondents rated palliative services with the best possible response. |

*(Continued)*

**Table 4.** (Continued)

| Author(s), year | Development/Background & pilot-tested | Content | Teaching strategies | Evaluation (Outcomes, measures and time points) | Main results* |
|---|---|---|---|---|---|
| Quinn et al., 2008 | Developed by<br>• evidence from a literature review (not specified);<br>• collective clinical experience of project team<br>• Australian Standards for Palliative Care<br>• input from an expert panel/advisory group (n = 13) comprising academics, policy makers, clinicians, caregiver representatives representing clinical and academic palliative care fields;<br>Pilot-tested: NR | Program Day 1:<br>1. Clinical and public health approaches to palliative care<br>2. Decision making in palliative care: ethical and legal challenges<br>3. Assessing and responding to spiritual and cultural issues<br>4. Family centered care<br>5. Grief, loss, and bereavement<br>Program Day 2:<br>6. Communicating with patients, families and colleagues<br>7. Palliative care and nonmalignant disease<br>8. (doctors and nurses) frequently asked questions OR Multidisciplinary Team- does it work? (allied health clinicians)<br>9. Care of an imminently dying person and their family<br>Panel discussion | Didactic, question and answer sessions, workshops and panel/case discussions; Resources, appropriate to each session, including relevant reading materials, websites, and reference lists were provided to each participant.<br>Closing session in form of a panel discussion including local palliative care services.<br>Facilitators and their professional background:<br>Sessions presented by specialist palliative care clinicians from a range of disciplines, but not further specified; | a.) *Demographic survey*<br>b.) *Generic palliative care questionnaire–* measuring participant perception related to importance, knowledge, and confidence with palliative care in eight areas (self-developed, 61 items on a 5-point Likert scale), pre-and post-program participation + additionally 1-month follow-up<br>c.) *Session Evaluation Questionnaire–* measuring any benefit to level of interest, extent of new information learned and perceived usefulness of each session (3 questions, 4-point Likert scale)<br>d.) *Total program evaluation* (self-developed, qualitative evaluation) after the program with open-ended questions to encourage participants to provide comments not captured by questionnaire (e.g. strength and weaknesses of the program, ongoing learning needs, and identify challenges in implementing change in their workplace) *changes in daily practice and communication since participation*;) Focus group *1-month follow-up (with n = 8 participants)*: aimed to explore the organization, appropriateness and relevance of the program content to practice, recommendations for improvements and possible concerns or issues by participants<br>Follow-up: 1-month follow-up for *b.)*<br>Development: self-developed, not validated tools; | b.) Multivariate effects were found within groups for time [p < .0001], showing significant differences with positive effects on importance, knowledge, and confidence in all eight areas from pre- to post-test, with stronger association for knowledge and confidence [p < .001], weaker for importance [p < .01—p < .05] (pretest scores were already fairly high though), response rate was ≈70%;<br>Follow-up questionnaire had only a response rate of 17%; no additional analyses repeated;<br>c.) Overall, session evaluations were positive, Session 6 and 7 were scored highest.<br>d.) Reported strengths were passionate conviction of excellent speakers, well-balanced coverage of the program, take-home resources were highly valued, time allowed for questions.<br>Reported suggestions for improvements were including more group sharing and participation, maintaining interest despite participant fatigue towards the end of each day;<br>e.) results: strong recommendation to colleagues, applicable to daily practice, improved motivation and confidence to change own practice. |
| Semple et al., 2017 | Content was derived from experiences of an expert team, which included an oncology nurse specialist, a professor in cancer care, a cancer nurse researcher, a family support coordinator and a parent; additionally, findings of empirical studies;<br>Pilot-tested: NR | Education sessions on:<br>• the importance of communicating with children about their parent's cancer<br>• difficulties reported by oncology professionals when working with parents;<br>• the essential need to support parents diagnosed with cancer;<br>• how to start the conversation with parents;<br>• guiding principles when talking to children about parental cancer<br>• Finding the words, using a case example affected by parental cancer<br>• Helpful tips when supporting families<br>Discussion round to reflect and share experiences | Delivery incorporated aspects of advanced communication skills training and learning methods to encourage reflection by participants and integration of case studies.<br>Facilitators and their professional background:<br>Two facilitators: one being a family support coordinator for Cancer Focus Northern Ireland, a cancer charity that provides special support to families with parental cancer; the other being an experienced clinical nurse specialist leading a program of work on family centered cancer care; | Self-generated survey including single items and a free text section for comments;<br>a.) Demographic background<br>b.) Perceived confidence and competence to communicate with parents about parental cancer (three clinical scenario-based single item questions, 10-point Likert scale); *pre-post participation*;<br>c.) Perceived increase of knowledge, recommendation to colleagues, influence on daily practice post-participation (three single item questions, 10-point Likert scale)<br>d.) Free-text section<br>Follow-up: none Development: self-developed, development by experienced researchers and extensively described and each item was reviewed and refined for content and face validity, pilot-tested with a small group of HCPs; | b.) significant improvements in participants' perceived confidence and competence after the educational session on all three clinical scenario-based questions [p < .001]; improvements did not differ by professional background, years of experience, or having received formal training before; there were some significant group differences at baseline (nursing students vs. medical staff), however all participants' confidence and competence increased in a similar way;<br>c.) participants perceived the education session increased their knowledge, that it would influence their daily practice and they would recommend it to colleagues; only statistically significance difference between mean scores for staff nurses compared to nursing students on knowledge gained and recommendation of training to others [p < .001]; statistically significant lower mean scores for allied health professionals influence on daily practice [p < .001].<br>d.) themes within free text sections were increased knowledge, useful for practice, optimal delivery to enhance learning, desire for more in-depth training; overall very positive responses and feedback. |

(*Continued*)

**Table 4.** (Continued)

| Author(s), year | Development/Background & pilot-tested | Content | Teaching strategies | Evaluation (Outcomes, measures and time points) | Main results* |
|---|---|---|---|---|---|
| Turner et al., 2009 | • literature review on key issues: the impact of advanced cancer on parents and children; strategies to promote adjustment; communication with patients with cancer, staff stress, burnout and resilience research<br><br>• Focus groups with oncology nurses to identify educational needs<br><br>• Expert panel (n = 6 nurses with a national profile in nursing education and service delivery) reviewed the manual, documenting their opinions about the relevance, style and content of each section of the manual, noting if any important topics were missing;<br><br>Pilot-tested: only the manual by expert panel | Self-directed educational manual including three modules: overview of evidence about communication in oncology, with special reference to parents with advanced cancer; Needs of children of parents with cancer; Issues for parents with advanced cancer; Interventions to promote adjustment in children; prompts and suggestions to be used in response to specific challenges, such as parents who express anger, guilt about their illness.<br><br>Communication skill training workshop including overview of the evidence about communication in oncology with special reference to parents with advanced cancer, after which participants' developed role-plays and participated in these. | Presentations, group work to develop role-plays (participants developed role-plays on their own), participate in role-plays, constructive feedback for role-plays<br>Participants had to define their own learning needs, participate in role-plays and receive constructive feedback.<br>Facilitators and their professional background:<br>NR | a.) Demographic background<br>b.) Measures of burnout and psychological morbidity:<br>Maslach Burnout Inventory (MBI)<br>General Health Questionnaire 28 (GHQ)<br>c.) Measures of perceived stress (4 items), confidence (5 items) and attitudes (5 items; all self-developed measure, pilot-tested:<br>5-point Likert scale)<br>d.) Assessment of knowledge–two clinical vignettes<br>e.) Assessment of skills–simulated 5-min video-taped interviews with simulated patient incl. quality rating (General Interactional skills) and responses to Scripted Cues<br>f.) Subgroup Analyses<br>g.) Acceptability survey 6-months after workshop participation (usefulness of components of the educational manual on a 5-point Likert scale; description of own emotional responses to the workshop, impact of the training on clinical practice)<br>Follow-up: 6-months after completion of post-questionnaire<br>Development: Mix of validated questionnaires (MBI, GHQ) and self-developed tools, being developed with literature review and results of focus groups (c) and partly pilot-tested (c) | b.) GHQ: significant reduction in Somatic subscale scores over time [p = .02];<br>c.) After participation nurses were significantly more likely to report taking an active role in 'caring for myself emotionally and spiritually'.<br>Significant increases in confidence about ability to provide support [p < .001], and information [p < .001], and to raise discussions about emotional issues with parents [p < .001];<br>d.) Over time there were significant reductions in pragmatic responses to the two vignettes [p = .001; p = .002],<br>recommendation of referral [p = .02; p = .03], and significant increases in promoting coping responses [p = .02; p≤.001].<br>e.) Comparisons revealed significant improvements in all categories of communication [p≤.001 till p = .002] except for responses to Unscripted Emotional Cues. Significant reduction in frequency of blocking responses [p≤.001].<br>f.) Small sample size, therefore limited number of subgroup analyses conducted. Nurses scoring high on Emotional Exhaustion at recruitment were significantly more likely to have low confidence in ability to provide support for parents [p = .015], and a low degree of initiative in self-care strategies [p = .013]. Nurses aged ≤ 40 years had significantly higher worries about what to say when patients were distressed [p = .041], but there were no other significant associations between younger age and confidence or attitudes. Nurses with previous experienced bereavement were significantly less likely to score as 'cases' on the GHQ [p = .012].<br>g.) Response of N = 17 nurses only. Training was generally acceptable. Qualitative analysis of responses revealed 2 dominant themes: increased confidence and recognition about support does not involve solving patient's problems. |

*(Continued)*

**Table 4.** (Continued)

| Author(s), year | Development/Background & pilot-tested | Content | Teaching strategies | Evaluation (Outcomes, measures and time points) | Main results* |
|---|---|---|---|---|---|
| Wittenberg et al. 2020 | • Modified and developed based on Family Caregiver Communication Typology (FFCT), grounded in communication theory and included evidence-based communication skills. <br>• Face validity to typology was given by feedback from 2-day COMFORT Communication training course participating nurses. <br>• Revision of module by five nurse educators working range of institutions (during early development phase, and after module completion) including open feedback on content, design and sequence of materials. <br>• An instructional education designer proficient in online learning platforms and educational theory then designed the online format of the educational material. Pilot-tested: yes, with five nurse educators | Modified F module COMFORT including three learning outcomes (knowing the four different caregiver communication types, describing the family communication patterns for each caregiver type, responding to each given scenario in a manner responsive to different caregiver types) within 9 Sections | Online training including videos, case study examples, reflection and assessment exercises, feedback from the instructor on participants' submitted reflection. Included all of the American Association of College of Nurses curriculum guidelines required for baccalaureate nurses including practice-based learning and improvement, evidence-based practice, interprofessional and interpersonal communication skills, professionalism, and system-based practice Facilitators and their professional background: <br>N/A | Pre-post survey immediately prior to and immediately after participation assessing: <br>a.) Demographic survey (baseline only) <br>b.) attitude (2 items) <br>c.) knowledge (5 items) <br>d.) behavior (3 items) <br>e.) open-ended questions with section 5–8 capturing knowledge and behavior (3 questions) <br>Follow-up: none <br>Development: developed by research team based on prior communication research; | b.)–d.) significant statistical increase in the scores from pre-test to post-test [p <.001], therefore online module had an effect on nursing student attitude, knowledge, and behavior for communicating with caregivers (with first- and fourth-year students showing biggest increase across these 3 topics. <br>e.) majority of responses correctly described the family communication pattern presented in the case study: Descriptions of their behavior with each caregiver type in response to a case study scenario revealed that student mastery of content (level 2 or higher) ranged from 40%‒56% across caregiver types. Responses classified at levels 2–4 demonstrate fluidity of knowledge skills in addition to goal complexity and mastery. |
| Wittenberg et al. 2021 | 3-part communication training webinar series built and offered based on COMFORT Curriculum, adapted by reviewing literature on communication challenges in Nairobi area/Africa; Pilot-tested: NR | Adapted COMFORT curriculum for a Webinar-Based Training including 3 of COMFORT's 7 modules: C-Connect, O-Options, F-Family caregivers; applied as a 3-part webinar series; | Webinar included chat feature, recording, slide decks, learning objectives, interactive polling, and application of exercises for audience members to practice new communication skills | Online survey assessing: <br>a.) Demographics <br>b.) Previous educational training on specific communication topics <br>c.) Pre-post assessment of adapted (?) Comfort with Communication in Palliative and End of Life Care (C-COPE) (15 items reported, 5-point Likert scale <br>d.) Survey on education program (3 items for each webinar, 5-point Likert scale) and overall course evaluation (? Items, 5-point Likert scale) <br>Follow-up: none <br>Development: NR and not possible to follow adapted C-COPE assessment; | c.) significant change in level of perceived communication comfort in 75% of participants (N = 9, p-value not reported); <br>d.) overall webinars received high evaluation (M 4.5 from 5), Module F on Family Caregivers ranking highest (M 4.65 from 5) |

* only significant findings will be reported due to clarity of table

CST = Communication skills training; Comskil program: developed by Kissane et al. at the Memorial-Sloan-Kettering Cancer Center (MSKCC) in the USA; NR = not reported; SPAs: standardized patient assessments; COMFORT TM SM Communication Curriculum developed by Elaine Wittenberg, PhD and Joy Goldsmith, PhD originally as a 2-day face-to-face training for the palliative care setting; EOL: end of life; HCPs = Healthcare professionals; NA = not applicable/applied.

COMFORT curriculum without further information or explanation for possible variations in content of their CST within their reports. Fuoto et al. [34] described their module as "Family module: support caregiver involvement and understanding" and Wittenberg et al. (2021) [37] "F-Family caregivers".

## Didactic techniques/materials

All included studies combined various didactic techniques and materials within their training program. Role-play exercises with regular feedback were part of five studies [23, 31–34], role-play exercises with simulated patients were incorporated in three studies [23, 31, 32]. All studies but the study using the e-learning [35] gave some kind of presentation (e.g., power-point introduction on training or overview of communication skills). Discussion rounds were part of the training in four studies [33, 34, 36, 38] and videos e.g., to illustrate key skills or family needs included four studies [31–33, 35]. Moreover, various studies used written material in form of manuals [23, 36], booklets [31, 32] or pocket-cards [34]. Five studies reported on professional background of CST facilitators [31–33, 36, 38], which varied greatly between studies (for details see Table 4).

## Outcome measurement

Included studies varied considerably in defined outcomes and applied instruments (e.g., number of items, scales, description of adapted instruments). Most instruments have been self-developed without validation (see Table 4 for details). Two studies applied Kirkpatrick's framework [29] for training evaluation, focusing on the first two levels: participant's reaction and learning [31, 32].

*Participants' satisfaction* with the CST was assessed post-training participation. Five studies evaluated satisfaction using quantitative evaluation surveys [31–33, 36, 37] and two qualitative methods (e.g., open-ended questions [35]; focus groups [36]). Some studies assessed overall satisfaction with CST [33, 36, 37], others assessed satisfaction with individual sessions/ modules [31, 32, 37, 38]. Overall, assessment of participants' satisfaction varied considerably.

Majority of studies (n = 8) included a pre-post participation assessment of *self-efficacy and/or perceived confidence in communication competencies* [23, 31–34, 36–38]. Three studies analyzed change of HCPs' *attitudes* [23, 33, 35] (e.g., towards the importance of learned skills [33]), three studies analyzed change of HCPs' self-perceived *communication behavior* in daily practice [34, 35, 38] and three *observed communication skills* assessed through simulated patient assessments (SPAs) [23, 31, 32] pre-post training participation. Two studies measured change in *perceived importance* of communication [33, 36]. Four studies assessed change of *knowledge* how to support parents and families [23, 35, 36, 38], e.g., knowledge on palliative care [36] or retrospectively perceived increase of knowledge on supportive needs of parents and families [38]. HCPs' *general health* (burnout and perceived stress as secondary outcomes) [23] and *patient-reported outcomes* [34] (adapted version of the patient-family satisfaction with End-of-Live care survey (FEPC), a post-death survey for relatives originally developed by the Natioanl Hospice and Palliatve care Organization in Virginia, USA, however not available) were each reported in one study.

## Evaluation of CST following Kirkpatrick's framework for training evaluation

*Reaction–Participant's satisfaction with the training*. Overall, participants' reaction to CST was predominantly positive. Participants rated the trainings beneficial for applying it to their daily practice [36, 38], to increase their confidence [23] and would recommend it to their colleagues

[36, 38]. Reported suggestions were e.g., increasing group sharing exercises [36] or discussion/exchange rounds to share their experiences with affected families [33].

*Learning–Effects on participant's communication confidence, attitudes and knowledge.* Statistically significant improvement on participants' self-reported ***self-efficacy in communication competencies*** were found in seven studies [23, 31, 32, 34, 36–38] with considerable variation in defined outcomes and applied instruments (see Table 4 for details). One study did not report detailed statistic parameters [37]. Two of the three studies assessing participants' ***attitudes*** reported significant improvements over time [23, 35]. Only one of the two studies assessing ***perceived importance*** of communication found significant improvements over time [36]. Regarding ***knowledge***, three studies reported significant improvements over time [23, 35, 36], with one study missing clear and detailed statistic parameters [35].

*Behavior–Participant's change in behavior.* Of the three studies assessing daily ***communication behavior***, only one study reported on significant changes, but did not provide statistic parameters [35]. Semple et al. assessed change of communication behavior only at post-participation without comparison over time [38] and Fuoto et al. with an open-answer format only [36]. Significant changes in ***observed communication skills*** were found in three studies. Banerjee et al. and Cannity et al. reported significant improvements for overall skills using both the same Comskil coding manual [31, 32], Turner et al. for five of their six categories on measuring General Interaction skills and responses to Scripted Cues [23].

*Results–other improvements in patient-oriented healthcare.* One study assessed participants' ***general health*** using the General Health Questionnaire 28 (GHQ), the level of perceived stress (self-administered) and burnout with the Maslach Burnout Inventory (MBI) [23]. There were no significant changes in stress and burnout or level of perceived stress. Significant decrease in the somatic subscale of the GHQ was reported. Regarding the patient-reported outcomes measuring patient-family satisfaction with care no significant differences between pre- and post-training scores were found [34].

Both studies specifically focusing their CST to provide support for cancer patients parenting minor children found significant changes within the pre-and post training assessment for multiple outcomes [23, 38] (see Table 4).

## Methodological quality assessment

The methodological quality of included studies was rated as "fair" in six [32–36, 38], "poor" in two [31, 37] and "good" in only one included study [23] (see Table 3). None of the included studies reported on a sample size calculation, the statistical methods of two studies were of poor reporting quality [31, 37], the eligibility criteria for participants were only partly or not described in eight studies [23, 31–33, 35, 38], outcome measures were not or only partly reported in all studies, and only two studies reported on consistent delivery of intervention [23, 38].

## Discussion

This review aimed to provide an overview of existing CST interventions for HCPs in oncology explicitly addressing child- and parent-specific aspects in adult cancer care. Second, the review aimed to assess reported outcome measures associated with the CST's evaluation. The third aim was to report on CST effectiveness. Since only two studies were identified explicitly reporting on a CST solely focusing on parental cancer, we broadened our focus during the screening process to also include studies reporting on a family-specific module within their CST. Thus, in total, we included nine studies with at least one module on child- or family-specific aspects in communication in cancer care. The seven included studies including a family-specific

module did not provide details what is included (e.g., parental-specific aspects during cancer care). Hence, it remains unclear if and to which extend children as relatives of cancer patients are explicitly addressed. Findings of the present work are consistent with previous research identifying a lack of communication skills trainings in oncological care especially for HCPs caring for patients experiencing additional burden and needs [41].

In our included studies, nurses represented a large proportion of participants with six studies including nurses only [23, 31–35] and two studies mainly including nurses [36, 38]. This is not surprising as one frequently evaluated CST is the COMFORT curriculum explicitly developed for nurses [40]. As nurses spend a considerable amount of their working time caring for patients, developing a close relationship with their patients and relatives [42], they are often confronted with patient's specific needs and provide emotional support [43]. Additionally, shortage of nursing staff globally and a continuous physically and emotionally draining job [23, 44] increase the need to enhance effective communication with patients and their families to reduce stress experience and emotional exhaustion in nursing profession [43, 45, 46].

Physicians usually are the key contact and person of trust for patients during cancer care [47]. Therefore, they can act as gatekeepers for additional support according to child- and family-related needs. However, in the included studies only few physicians participated.

Studies on child- and parental-related issues report lack of knowledge and specific communication skills as well as perceived limited competence on parental issues in clinicians in cancer care [10, 11, 21]. This strongly indicates a need for 1) specifically developed training programs for physicians and oncologists incorporating child- and parent-specific aspects or 2) optimization of access to existing interventions to improve participation of physicians, e.g., by including incentives or adapting trainings to their specific needs and working schedule.

Six of the included studies found significant improvements in either self-efficacy and/or confidence, behavior and knowledge for general communication skills, two additional studies for specific communication aspects in parental cancer. This implies that CSTs are a promising approach to improve HCPs communication skills including specific skills on parental cancer and support building a bridge to communicate effectively with affected parents and their families. This implication is supported by previous research, indicating increased self-efficacy, knowledge and skills [48] will in turn improve (a) HCP's communication behavior, (b) HCP's satisfaction with communication and their mental well-being health (e.g., reduced emotional burn-out) [25], and (c) outcomes for patients and their families (e.g., reduced stress and feelings of anxiety, improved satisfaction with care [26, 49]). However, findings are not generalizable due to small sample sizes in most studies included in this review and only two included studies applying a specific CST on parental cancer.

The overall methodological quality of included studies was fair to poor. Applied outcome measures varied considerably and psychometric properties of measures were insufficient. However, validated and reliable tools assessing specific communication skills and behavior in child- and family-specific aspects in cancer care are rare [23, 38, 50]. Hence, there is a need for rigorously developed and psychometrically sound instruments. Moreover, objective simulated patient assessments (SPAs) should be included in future studies as they are the gold standard for evaluation of CSTs [51, 52]. Clinical case vignettes, as used in one included study [23], have been found to be comparable to SPAs [52]. However, development of vignettes should be standardized and follow current recommendations [53].

## Study limitations

This study has several limitations. First, this systematic review focused on CSTs with a specific module on child- or family specific aspects in cancer care. Though our search strategy was

extensive, the articles reviewed may not represent all CSTs with such specific modules in cancer care given the restrictions of search terms used, databases searched and requirements for English- or German-language due to language restrictions of the authors. However, by including a thorough secondary literature search, additional relevant CSTs were included. Second, as included studies varied considerably in e.g., CST content and outcome assessment and tools used, comparison of CSTs and their quality of evidence is difficult and generalizability is impeded. Additionally, based on our quality assessment, only one study with good methodology design was included.

## Clinical implications

Overall, implication for future research is to develop a structured and theory-based communication skills intervention for HCPs in oncology to improve family-centered cancer care, specifically when a parent has cancer [38, 43]. Future studies should develop specific trainings to enhance HCPs communication skills, knowledge and self-efficacy to address child- and family-specific aspects when a parent has cancer. Also, these studies should provide an evaluation using state of the art methodology (e.g., including a control group thorough outcome assessment with validated, and pilot-tested outcome measurements based on e.g., Kirkpatrick's model of evaluation) [29, 50]. Additionally, newly developed interventions should specifically address physicians and oncologists and if possible be adapted to their needs to increase participation of this specific HCP group. Existing studies including a family-specific module should provide further detail on the topic of "family communication", e.g., if minor children are included as family members [29, 50].

## Conclusion

This systematic review gives an overview of existing CSTs for HCPs on parenthood and cancer. Despite a high need for a specific CST to improve HCP's communication skills regarding parental cancer, only two CSTs focusing on parental cancer were identified, the remaining seven studies only included a brief module on family communication. The quality of evidence for included studies remains insufficient. Due to the lack of specific CSTs and poor or only fair quality of the included studies, further CSTs on aspects of parental cancer should be developed and evaluated rigorously.

## Supporting information

**S1 Checklist. PRISMA 2009 checklist.**
(DOC)

## Author Contributions

**Conceptualization:** Wiebke Frerichs, Wiebke Geertz, Lene Marie Johannsen, Laura Inhestern, Corinna Bergelt.

**Data curation:** Wiebke Frerichs.

**Formal analysis:** Wiebke Frerichs, Wiebke Geertz, Laura Inhestern.

**Investigation:** Wiebke Frerichs.

**Methodology:** Wiebke Frerichs, Wiebke Geertz, Laura Inhestern.

**Project administration:** Wiebke Frerichs.

**Resources:** Wiebke Frerichs.

**Supervision:** Laura Inhestern, Corinna Bergelt.

**Validation:** Wiebke Frerichs, Wiebke Geertz, Lene Marie Johannsen.

**Writing – original draft:** Wiebke Frerichs.

**Writing – review & editing:** Wiebke Frerichs, Wiebke Geertz, Lene Marie Johannsen, Laura Inhestern, Corinna Bergelt.

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
