## [Decision Letter · Decision Letter 0]

2 Aug 2022

PONE-D-22-11030Child- and family-specific communication skills trainings for healthcare professionals caring for families with parental cancer: a systematic reviewPLOS ONE

Dear Dr. Frerichs

Thank you for submitting your manuscript to PLOS ONE. After careful consideration, we feel that it has merit but does not fully meet PLOS ONE’s publication criteria as it currently stands. Therefore, we invite you to submit a revised version of the manuscript that addresses the points raised during the review process.

ACADEMIC EDITOR: Thank you for choosing Plos One to send the manuscript. According to the reviewers' evaluation, the manuscript is adequate, and only needs a few minor corrections for its acceptance. Therefore, I encourage authors to carefully review the points raised by the reviewers.

Please submit your revised manuscript by  Aug 15, 2022. If you will need more time than this to complete your revisions, please reply to this message or contact the journal office at plosone@plos.org. Please include the following items when submitting your revised manuscript:A rebuttal letter that responds to each point raised by the academic editor and reviewer(s). You should upload this letter as a separate file labeled 'Response to Reviewers'.A marked-up copy of your manuscript that highlights changes made to the original version. You should upload this as a separate file labeled 'Revised Manuscript with Track Changes'.An unmarked version of your revised paper without tracked changes. You should upload this as a separate file labeled 'Manuscript'.If applicable, we recommend that you deposit your laboratory protocols in protocols.io to enhance the reproducibility of your results. Protocols.io assigns your protocol its own identifier (DOI) so that it can be cited independently in the future. For instructions see: https://journals.plos.org/plosone/s/submission-guidelines#loc-laboratory-protocols. Additionally, PLOS ONE offers an option for publishing peer-reviewed Lab Protocol articles, which describe protocols hosted on protocols.io. Read more information on sharing protocols at https://plos.org/protocols?utm_medium=editorial-email&utm_source=authorletters&utm_campaign=protocols.

We look forward to receiving your revised manuscript.

Kind regards,

Manoelito Ferreira Silva Junior, Ph.D.

Academic Editor

PLOS ONE

Journal Requirements:

Additional Editor Comments (if provided):

Thank you for choosing Plos One to send the manuscript. According to the reviewers' evaluation, the manuscript is adequate, and only needs a few minor corrections for its acceptance. Therefore, I encourage authors to carefully review the points raised by the reviewers.

Reviewers' comments:

Reviewer's Responses to Questions

**Comments to the Author**

1. Is the manuscript technically sound, and do the data support the conclusions?

Reviewer #1: Partly

Reviewer #2: Yes

2. Has the statistical analysis been performed appropriately and rigorously? 

Reviewer #1: Yes

Reviewer #2: Yes

3. Have the authors made all data underlying the findings in their manuscript fully available?

Reviewer #1: No

Reviewer #2: Yes

4. Is the manuscript presented in an intelligible fashion and written in standard English?

Reviewer #1: Yes

Reviewer #2: Yes

5. Review Comments to the Author

Reviewer #1: The study presents an adequate methodological structure. Some recommendations will be made to better adapt the proposal and understand the development of the article.

1. It is suggested to update the databases. The last update is more than 6 months old.

2. Justify the language restriction. It was important to consider it as a limitation of the study.

3. Justify the exclusion of other types of quantitative studies. Including justifying the absence of a control group, which allows better comparability of training results.

In the results, 2 studies include students in the sample. It is important to point out if the results are treated differently in the studies, in relation to the professionals.

In table 4, some numerical data can be inserted into the results for better visualization of the data presented.

The review brings only 2 studies that address the relationship between parents and children. Most other articles focus on general family communication guidelines.

Thus, I understand that the analyzed articles extrapolate the objective "a" proposed in the review. Can be reviewed.

In addition, the Introduction directs toward understanding the relationship. Depending on the targeting of the objective and the results, the text needs to be reorganized, expanding the justification for family communication. The analysis of studies that consider the relationship between parents and children may be highlighted in a sub-analysis.

The clinical implications should consider the importance of the study for training that considers the family relationships of parents and children in the oncological context, which is the central objective of the presented study.

Reviewer #2: The topic researched by this article (Communication skills trainings (CST) in oncology) is extremely relevant for the scientific community and for healthcare professionals.

The authors meet the PLOS criteria for publication and responded to the objectives proposed in the research.

6. PLOS authors have the option to publish the peer review history of their article (what does this mean?). If published, this will include your full peer review and any attached files.

Reviewer #1: No

Reviewer #2: No

---

## [Author Response · Author response to Decision Letter 0]

2 Sep 2022

Editor comments:

1. Thank you for choosing Plos One to send the manuscript. According to the reviewers' evaluation, the manuscript is adequate, and only needs a few minor corrections for its acceptance. Therefore, I encourage authors to carefully review the points raised by the reviewers.

Thank you for the opportunity to review and resubmit our manuscript. We carefully revised the manuscript according to the reviewer’s comments and also revised the Abstract, as word count was exceeded. We hope that it is now considered appropriate for publication in PLOS ONE.

Response to reviewer comments

Reviewer #1: The study presents an adequate methodological structure. Some recommendations will be made to better adapt the proposal and understand the development of the article.

Thank you for your valuable peer review. 

1. It is suggested to update the databases. The last update is more than 6 months old.

Thank you for pointing this out. As suggested, we did an update of the search within the databases on August 12th and updated “Fig 1. PRISMA flow diagram for systematic reviews” as well as the Methods section accordingly. In summary, a total of n=1455 additional records were identified, of which n=713 screened for title abstract screening, and of which n=2 full-text articles were assessed for eligibility. N=0 articles met inclusion criteria and therefore no additional articles were included in the qualitative synthesis. 

2. Justify the language restriction. It was important to consider it as a limitation of the study.

Thank you for pointing this out. We have revised the manuscript and included the language restriction of the authors within the Methods - Eligibility section (“Due to language restriction of the atuhors, peer reviewed …”) and further pointed it out as a study limitation within the discussion section (“... requirements for English- or German-language due to language restrictions of the authors.”) 

3. Justify the exclusion of other types of quantitative studies. Including justifying the absence of a control group, which allows better comparability of training results.

Thank you for raising this concern. We did not exclude other types of quantitative studies and would have included studies incorporating a control group, if there were any identified within our search. We agree, that our description can benefit from more clarity and therefore have added a sentence within the Methods – Eligibility criteria and study selection section “… reporting any type of CST with a pre-post design (e.g., single arm intervention studies or studies including a control group) regarding outcomes … “. Further, we clarified the section Description of included studies by adding following information “Included studies used a quasi-experimental design with pre-post measurement only and no studies were identified including a control-group.”

4. In the results, 2 studies include students in the sample. It is important to point out if the results are treated differently in the studies, in relation to the professionals.

Thank you for raising this concern. As in these two studies the sample consisted of students only, the results could not be treated differently. However, we realized that our description of the two studies entailing students only was limited and therefore revised the manuscript to make it more clear: “Most studies included qualified HCPs (23, 31, 33-36, 38), two studies included nursing students only (32, 35) (28, 31) (see Table 3)”. 

5. In table 4, some numerical data can be inserted into the results for better visualization of the data presented.

Thank you for this suggestion. We agree that the visualization of the data presented so far was not optimal. Therefore, we added significant numerical data to table 4 to aid understanding of presented data of the following authors: 

o Banerjee et al 2017 regarding outcomes on participants learning; 

o Cannity et al 2021 regarding outcomes on participant learning; 

o Fuoto & Turner 2019 regarding outcomes on communication confidence, communication satisfaction;

o Quinn et al 2008 regarding outcomes on Generic Palliative Care questionnaire; 

o Semple et al 2017 regarding outcomes on perceived confidence and competence to communication;

o Turner et al 2009 regarding outcomes on General Health Questionnaire, perceived stress, attitudes and confidence, clinical vignettes, simulated patient interviews and subgroup analyses; 

o Wittenberg et al 2020 regarding outcome one numerical data on pre- post-test on attitudes, knowledge and behavior; however, it is not clear how this was analyzed and no overview (e.g., table) of data is provided; 

6. The review brings only 2 studies that address the relationship between parents and children. Most other articles focus on general family communication guidelines.

Thus, I understand that the analyzed articles extrapolate the objective "a" proposed in the review. Can be reviewed.

Thank you very much for this important point. During the search process only two studies focused on child-and parent-specific aspects only. As the remaining seven studies had at least a family-specific module and were identified through our search strategy (which entailed a search term “famil”), but did not further specify the content of the family-module, e.g., if communication with minor children of cancer patients were included within the module, we decided to broaden the scope to child-and family-specific communication skills trainings. But, we agree, that the objective is misleading and have therefore revised the manuscript as described below. 

1. Revised the objective “a” accordingly in the INTRODUCTION section: 

 “a.) provide an overview of existing CSTs for HCPs working in oncology addressing child- and parent-specific aspects in cancer care,”

2. Have revised the METHOD section why we also include family-specific communication trainings.

- “However, despite our extensive search strategy only two studies were identified during the study selection process to focus on child- and parent-specific aspects within their CSTs. Therefore, we decided to broaden the focus of this systematic review and to include studies, which entail a child- and family-specific module within their CST.” 

- And included within the Table 2. Inclusion and exclusion criteria the criteria: Studies evaluating a communication training or educational program including at least a module on child-, parent- or family-specific themes;

3. Revised the RESULT section accordingly to highlight the results of the content of the training

- by highlighting the two specific trainings on parents with cancer at the beginning of the RESULT section: “The main literature search identified two studies specifically addressed the subject of cancer patients parenting minor children within their CST and five studies incorporated a brief family module within their CST. The first update added another two studies evaluating a CST for HCPs in oncology, including a brief module on family-specific aspects in cancer care. In total, nine studies were included in this review (Fig 1)

- By addressing the significant results of the two specific trainings on parental cancer: “Both studies specifically focusing their CST to provide support for cancer patients parenting minor children found significant changes within the pre-and post training assessment for multiple outcomes (23, 38) (see Table 4).”

4. Revised the Discussion section accordingly to make the objective and results of the review clear:

“This review aimed to provide an overview of existing CST interventions for HCPs in oncology explicitly addressing child- and parent-specific aspects in adult cancer care. Second, the review aimed to assess reported outcome measures associated with the CST’s evaluation. The third aim was to report on CST effectiveness. Since only two studies were identified explicitly reporting on a CST solely focusing on parental cancer, we broadened our focus during the screening process to also include studies reporting on a family-specific module within their CST. Thus, in total, we included nine studies with at least one module on child- or family-specific aspects in communication in cancer care. The seven included studies including a family-specific module did not provide details what is included (e.g., parental-specific aspects during cancer care).”

7. In addition, the Introduction directs toward understanding the relationship. Depending on the targeting of the objective and the results, the text needs to be reorganized, expanding the justification for family communication. The analysis of studies that consider the relationship between parents and children may be highlighted in a sub-analysis.

Thank you for this valuable comment. We agree and have adjusted the introduction by expanding to family communication within cancer care including applicable references. We hope in combination with adjusting comment No 6, these changes lead to a clearer understanding of the importance of family communication within cancer care, specifically when cancer patients parent minor children. 

Adjustments to the INTRODUCTION: 

“In order to provide high-quality, patient-centred cancer care, involvement of family and their specific needs is essential (15, 16). Family members are often the primary support for cancer patients (17) and act as caregiver and thus are impacted by cancer as well (18). As family communication is associated with relationship functioning and adjustment to the cancer diagnosis (18), it is essential for HCPs to provide support to cancer patients and their families on family communication issues, e.g., open communication. In order to identify potential cancer patients parenting minor children, it is key to know about the patient’s family status and if applicable to proactively address child- and family specific themes within cancer care.”

8. The clinical implications should consider the importance of the study for training that considers the family relationships of parents and children in the oncological context, which is the central objective of the presented study.

Thank you for this important feedback. We have adjusted the CLINICAL IMPLICATION section accordingly by including a specific focus on family communication when cancer patient parent minor children and regrouped the argumentation.

“Overall, implication for future research is to develop a structured and theory-based communication skills intervention for HCPs in oncology to improve family-centered cancer care, specifically when a parent has cancer (38, 43). Future studies should develop specific trainings to enhance HCPs communication skills, knowledge and self-efficacy to address child- and family-specific aspects when a parent has cancer. Also, these studies should provide an evaluation using state of the art methodology (e.g., including a control group thorough outcome assessment with validated, and pilot-tested outcome measurements based on e.g., Kirkpatrick’s model of evaluation) (29, 50). Additionally, newly developed interventions should specifically address physicians and oncologists and if possible be adapted to their needs to increase participation of this specific HCP group. Existing studies including a family-specific module should provide further detail on the topic of “family communication”, e.g., if minor children are included as family members. (29, 50).”

Reviewer #2: The topic researched by this article (Communication skills trainings (CST) in oncology) is extremely relevant for the scientific community and for healthcare professionals.

The authors meet the PLOS criteria for publication and responded to the objectives proposed in the research.

→ Thank you for your valuable peer review.

---

## [Editor Report · Decision Letter 1]

24 Oct 2022

Child- and family-specific communication skills trainings for healthcare professionals caring for families with parental cancer: a systematic review

PONE-D-22-11030R1

Dear Dr. Wiebke Frerichs

We’re pleased to inform you that your manuscript has been judged scientifically suitable for publication and will be formally accepted for publication once it meets all outstanding technical requirements.

Kind regards,

Manoelito Ferreira Silva Junior, Ph.D.

Academic Editor

PLOS ONE

Additional Editor Comments:

I thank the authors for the effort to answer each of the points raised by the evaluators.

Therefore, the items included in the final version of the file, and the answers included in the authors' letter, I inform you that the decision is to approve in the current format.

Best Regards.